# Study on the Strength Performance of Recycled Aggregate Concrete with Different Ages under Direct Shearing

**DOI:** 10.3390/ma14092312

**Published:** 2021-04-29

**Authors:** Xin Liang, Fang Yan, Yuliang Chen, Huiqin Wu, Peihuan Ye, Yuhuan Mo

**Affiliations:** 1Faculty of Geosciences and Environmental Engineering, Southwest Jiaotong University, Chengdu 611731, China; liangxin_gxust@163.com; 2School of Civil and Architectural Engineering, Guangxi University of Science and Technology, Liuzhou 545006, China; yf0396@foxmail.com (F.Y.); whq6329@163.com (H.W.); yepeihuan@yeah.net (P.Y.); yhmo0774@foxmail.com (Y.M.); 3Key Laboratory of Disaster Prevention and Structural Safety of Ministry of Education, Guangxi University, Nanning 530004, China

**Keywords:** recycled coarse concrete, direct shear test, early age, direct shear strength, residual strength

## Abstract

In order to study the mechanical properties of recycled aggregate concrete (RAC) at different ages, 264 standard cubes were designed to test its direct shear strength and cube compressive strength while considering the parameters of age and recycled aggregate replacement ratio. The failure pattern and load–displacement curve of specimens at direct shearing were obtained; the direct shear strength and residual shear strength were extracted from the load–displacement curves. Experimental results indicate that the influence of the replacement ratio for the front and side cracks of RAC is insignificant, with the former being straight and the latter relatively convoluted. At the age of three days, the damaged interface between aggregate and mortar is almost completely responsible for concrete failure; in addition to the damage of coarse aggregates, aggregate failure is also an important factor in concrete failure at other ages. The load–displacement curve of RAC at direct shearing can be divided into elasticity, elastoplasticity, plasticity, and stabilization stages. The brittleness of concrete decreases with its age, which is reflected in the gradual shortening of the elastoplastic stage. At 28 days of age, the peak direct shear force increases with the replacement ratio, while the trend is opposite at ages of 3 days, 7 days, and 14 days, respectively. The residual strength of RAC decreases inversely to the replacement ratio, with the rate of decline growing over time. A two-parameter RAC direct shear strength calculation formula was established based on the analysis of age and replacement rate to peak shear force of RAC. The relationship between cube compressive strength and direct shear strength of recycled concrete at various ages was investigated.

## 1. Introduction

Humans have produced a lot of pollution in the process of developing nature, such as contaminated material, construction rubbish. How to use them effectively to achieve energy saving and emission reduction has become an important issue. Recycled aggregate concrete refers to concrete made by replacing the natural aggregate content with an equal amount of recycled aggregate made from construction rubble. This technique can effectively alleviate the problems of construction rubble disposal, and it also promotes the protection of resources through recycling. In order to better apply, many researchers have carried out tests on the basic mechanical properties [1,2,3], cyclic mechanical properties [4,5,6], multiaxial mechanical properties [7,8,9,10], and other features of RAC to upscale its application. In addition, many scholars [11,12] have also conducted research on other waste utilization.

Shearing force is one of the basic mechanical properties of concrete, which is vital for structural design and finite element calculation. Various studies have aimed at describing the shear characteristics of concrete. For example, Zhang [13] obtained the stress–strain curve nearing the peak shear strength of natural aggregate concrete by conducting a shear test on short rectangular concrete beams. R. French [14] proposed a shear model with the aid of both static and dynamic direct shear tests on concrete materials. On the basis of investigating the shear mechanical properties and microstructure of three types of RAC, Liu [15] created stress–strain constitutive models of RAC under shear load and established formulas for its shear modulus. R.C.K. Wong [16] quantitatively analyzed the proportion of shear strength and delineated the shear strength envelope of concrete material under compressive shear load. By conducting a compressive shear test on RAC, Deng [17] discussed components of shear strength and proposed the shear strength formula of RAC. Yu [18] carried out experiments on plain concrete to explore its shear mechanical properties and proposed a corresponding compression–shear failure criterion based on shear tests on normal and high strength RAC. K. Rahal [19] found that the shear strength of RAC is lower than that of natural aggregate concrete (NAC), as measured by an indirect shear test. Tang [20] observed the influence of recycled aggregate on bonding behavior and mechanical properties of self-compacting concrete. Wang [21] considered that the deformation properties and direct shear strength of RAC are not as excellent as those of NAC. F. Ceia [22] studied the shear strength of the interface between RAC and NAC and found that the increase of replacement ratio is accompanied by shear strength decline. However, Gao [23] observed that the replacement ratio has a marginal effect on shear strength and shear deformation, which both increase nonlinearly with a rise in the replacement ratio. Yang [24] indicated that a slight change of the shear mechanical parameters is observed with the change of the replacement ratio at all temperatures. Waseem [25] concluded that the replacement ratio has a negligible impact on the shear strength of RAC and proposed a shear strength prediction model for RAC. Oldrich Sucharda [26] proposed a valid numerical modeling to analyze shear behaviors of reinforced concrete beams without shear reinforcement. In general, concrete shear tests have shown quite different results due to the varying shear test methods.

Concrete usually takes 28 days from pouring to reach working stability, but it is difficult to meet its maintenance conditions in real engineering situations. Speed and efficiency have become important in the concept development of modern construction engineering. The fast pace of building projects has led to the inevitable shortening of the concrete maintenance period; uncured concrete is often affected by intermittent or even continuous impacts from the environment. Therefore, it is often appropriate to shorten this curing period under conditions that ensure the bearing capacity of concrete and satisfy requirements for its regular use. Exploring the early strength of concrete is clearly of great importance for accelerating the progress of projects and for ensuring quality and safety. Jin-Keun Kim [27] associated mechanical properties with temperature and curing period and put forward relationships between each strength characteristic parameter. The compressive strength, elastic modulus, splitting tensile strength, and pullout strength of NAC were studied by H. S. LEW [28] while taking curing age into account. The compressive strength, as well as dynamic and static elastic modulus according to curing age, were explored by Han [29], who proposed the equation of the relationship between static elastic modulus and dynamic elastic modulus. Duy H. Nguyen [30] obtained tensile load–displacement curves of RAC at very early ages and revealed the law of direct tensile properties development for NAC. Mirian Velay-Lizancos [31,32] studied the E-modulus of NAC and RAC at an early age and discussed the influence factors of E-modulus. By analyzing the mechanical properties, including compressive strength, tensile splitting strength, modulus of elasticity, and shrinkage of RAC at early ages, Hu [33] indicated that the equations for these properties at early ages are not inconsistent between NAC and RAC.

At present, there are few studies on the shear performance of recycled concrete, and fewer studies on the direct shear performance of early age recycled concrete. Additionally, due to the difference in research methods, the test results are quite different. In order to provide a basis for the establishment of early age recycled concrete finite element model and engineering design, the present paper explores the fundamental shear properties of RAC at early ages. Relationships between shear strength and residual strength while considering the effects of replacement ratio and curing aging are analyzed based on experimental results.

## 2. Experimental Program

### 2.1. Materials

Ordinary Portland cement (P.O. 42.5, Guangxi Yufeng Cement Co., Ltd., Liuzhou, China) and tap water were adopted as binder and mixing water, respectively, and river sand was used as fine aggregate. Natural and recycled types were applied as coarse aggregate, the latter of which was crushed waste concrete from concrete specimens (C30-grade) for testing in the laboratory. The physical properties of coarse aggregate, as shown in Table 1, were taken from the literature [34].

The water content, water absorption, and crush index of RAC are significantly higher compared with NAC. However, NAC has a much larger bulk density and apparent density than RAC, which is mainly attributed to the existence of old mortar adhered around recycled coarse aggregate particles. In addition, recycled coarse aggregate has microcracks caused by crushing.

### 2.2. Specimen Design

For the experiment, concrete mixtures were designed to obtain C30 concrete following previous work [35]. Based on different concrete mixtures, RAC replaced NAC by weight (the replacement ratio was in the range of 0–100% with 10% difference), and additional water content was considered for RAC to compensate water absorption by recycled aggregates. A total of 11 types of mixtures were designed, as shown in Table 2.

Each mixture was shared into 12 specimens and divided into four groups for a direct shear strength test at the age of 3 days, 7 days, 14 days, and 28 days, respectively. Specimens were cube shaped with the size of 150 mm × 150 mm × 150 mm.

### 2.3. Test Setup and Method

The testing device consisted of an RMT-301 testing machine (Institute of Rock and Soil Mechanics, Chinese Academy of Sciences, Wuhan, China), upper shearing box, lower shearing box, rolling plate, and further components, as shown in Figure 1.

The bottom surface of the upper shearing box, the top surface of the lower shearing box, and the center of the lug were on the same plane as the middle of specimens. The right end of the upper shearing box was fixed by a circular arc-shaped self-adjusting lever with great rigidity, around which the upper shearing box could be rotated at a small angle. The self-adjusting lever was in contact with the restrictor plate that could slide vertically upward. The actuator had a spherical hinge end. The above structures ensured that the direction of maximum shear force in the specimen was kept unchanged during loading and eliminated the bending moment caused by potential eccentricity.

The testing device used in the concrete direct shear tests is presented in Figure 2.

After the specimen was installed, axial and lateral preloading were carried out to eliminate the gap between the specimen and the shear box. Axial loading was maintained in the preloading state (axial force < 1 kN, which is negligible, compared with lateral force) to compact the specimen, while lateral loading was subsequently applied through displacement control with a loading velocity of 0.02 mm/s until the specimen was ultimately damaged. The load–displacement curve of the full process in the direct shear test of the specimen was obtained through the acquisition system of the different devices. The direct shear loading diagrams obtained are shown in Figure 3.

## 3. Test Results

### 3.1. Failure Modes

The failure figures of specimens are presented in Figure 4, in which the figures are ordered by frontal, side, and failure planes of specimens. Both the friction area of the coarse aggregate interface and scraps were marked by a yellow ellipse, and the location of damaged coarse aggregate was marked by a red ellipse.

As shown in the figures, neat failure occurred in the frontal middle part of specimens under all conditions, showing a "one" shape. Although the crack inside was relatively convoluted, it was an overall straight line under all conditions. A gap occurred at the edges of specimens near the shear plane, which was caused by the shedding of aggregates at the corners. As a whole, front cracks and side cracks in specimens were not significantly affected by replacement ratio or age.

Two failure modes were proposed according to observations of the slag and the friction trace marked on the failure planes of specimens. One such mode includes the damage of interface between aggregate and mortar in a sample in which coarse aggregates were lacking clear damage, and the friction trace on the failure plane was obvious, which occurs at 3 days of age. Both the interface and coarse aggregate suffered damage in another sample, which occurred at 7 days, 14 days, and 28 days of age. Between 7 days to 28 days of age, coarse aggregates broke up into needles and flaky scrap, with the number of scrap pieces increasing with age.

It was concluded that age is the decisive factor leading to the kind of damage that occurs. The strength of coarse aggregate was constant, but the strength of the interface improved with age. When the coarse aggregate strength was greater than the interface strength, coarse aggregate separated from mortar once the interface was damaged, and the failure of specimens occurred simultaneously. When the interface strength was greater than or equal to the coarse aggregate strength, the irregular shape of coarse aggregate was rounded off first since the stress in the irregular sharp part of the coarse aggregate was relatively large; interface was then damaged, and the failure of specimens finally occurred.

### 3.2. Load–Displacement Curves

According to data on lateral force and lateral displacement recorded by the testing system, the load–displacement curves of specimens were drawn, as shown in Figure 5.

Typical load–displacement curves were divided into four stages, as shown in Figure 6, including the following:(1)Elasticity stage (OA): In this stage, the lateral force increased linearly with lateral displacement. This stage was steeper with age accounting for a rise in stiffness of specimens. The lateral force was offset by cohesive force between mortar and aggregate. According to data analysis, the limit of elasticity was 70% of the direct shear strength.(2)Elastoplasticity stage (AB): Lateral force increased nonlinearly with lateral displacement in this stage. The lateral force was offset by aggregate interlock force and cohesive force between mortar and aggregate. Once lateral force attained the limit of elasticity, microcracks occurred in the specimen with the rise in this force, and the excess force was counterbalanced by aggregate interlock force. Microcracks connecting to each other formed a main vertical crack when lateral force reached the peak shear force of the specimen, accounting for brittle failure. The elastoplasticity stage was longer with increasing age. In other words, specimen brittleness decreased relative to age, and the failure of specimen interfaces and specimens occurred almost simultaneously at 3 days.(3)Plasticity stage (BC): Lateral force decreased sharply with rising lateral displacement in this stage, which relationship was nonlinear. The lateral force was counterbalanced by aggregate interlock force and interface friction force. The aggregate was smashed, and the interlocking force decreased with increasing displacement, and the lateral force decreased at a slow rate until stabilization was attained.(4)Stabilization stage (CD): In this stage, the lateral force changed within 5% with increasing lateral displacement. All coarse aggregates were smashed, and the shear failure planes of specimens were flat. The lateral force was offset by interface friction force.

### 3.3. Characteristic Parameters

In order to study the strength characteristics of RAC under direct shearing, the peak shear force and the residual strength were extracted from the load–displacement curves, as shown in Table 3.

## 4. Discussions

### 4.1. Analysis of Peak Shear Force

There were considerable differences in the peak shear force of specimens at different ages and replacement ratios. The influence of age and replacement ratio on peak shear force was analyzed to understand the mechanical characteristics of RAC under direct shearing further.

#### 4.1.1. The Influence of Age

The peak shear force of specimens at different ages were compared, as shown in Figure 7a. The mean peak direct shear force and its rate of increase in specimens at ages of 3 days, 7 days, 14 days, and 28 days were 41.12 kN, 49.77 kN, 61.32 kN, and 73.53 kN, respectively. A rise in the shear force was observed with age at all replacement ratios.

The value of shear strength growth at each age, as a proportion of shear strength, is called the increase rate; this rate for different ages is shown in Figure 7b. Results show that the internal hydration reaction of forming concrete is violent in the early stage after casting, and the shear strength of the specimen increased rapidly. The growth rate of mean strength at 3 days is the largest, accounting for 55.9% of the peak direct shear force at 28 days. The intensity of hydration reaction decreases with age, and thus, the rate of increasing concrete strength drops. Accordingly, the mean direct shear strength of concrete at 7 days and 14 days reaches 67.79% and 83.4%, respectively, of its direct shear strength at 28 days.

#### 4.1.2. The Influence of Replacement Ratio

The comparison of the peak shear force of specimens with different replacement ratios is presented in Figure 8. As seen in the figure, the presence of recycled coarse aggregate has a significant impact on the peak shear of RAC.

In general, with the increase of the replacement ratios, the direct shear strength of 3 days, 7 days, 14 days recycled concrete increased, and the 28 days direct shear strength decreased. Values of mean peak direct shear force of RAC at ages of 3 days, 7 days, and 14 days, respectively, are 5.6%, 15.2%, and 14.6% higher than that of NAC. The effect of recycled coarse aggregate on the direct shear strength of RAC has mainly two aspects. On the one hand, the residual cement base attached to the surface of recycled coarse aggregate is beneficial to the growth of interface strength of RAC, thereby enhancing its overall shear strength. On the other hand, recycled coarse aggregates break up easier during the shearing process due to microcracks that form during the crushing of RAC, which is unfavorable to the strength of RAC. The shear strength of concrete is mainly affected by the interface strength, and the defect of coarse aggregate strength has little effect on shear strength at ages of 3 days, 7 days, and 14 days. Nonetheless, the mean direct shear strength of RAC reduces by 4.2% than that of NAC by the age of 28 days. This reversed trend is due to the fact that the strength defect of recycled coarse aggregate becomes the leading factor over time.

### 4.2. Analysis of Residual Strength

During the stabilization stage of the load–displacement curve, the lateral force changes slightly with increasing lateral displacement. In order to reduce the dispersion of data and obtain relatively true values of residual strength, the mean of lateral force difference was defined as within 5%, compared with the final lateral force as residual strength.

#### 4.2.1. The Influence of Age

The proportion of residual strength for RAC, compared with NAC at different ages, was calculated, as shown in Figure 9.

Residual strength is determined by interface friction, whose force is proportional to the interface friction coefficient since no axial pressure exists during the direct shear test. The friction coefficient is considered related to surface flatness and humidity in this study; hence, the increased values of these factors led to the reduction of the friction coefficient. Recycled coarse aggregates are more susceptible to shearing off due to strength defects, and the shear plane of RAC flattens with increasing replacement ratio. Furthermore, the additional water used in the mixture design of recycled concrete increases its humidity at early ages.

The residual strength of RAC is generally lower than that of NAC at 3 days of age. Additionally, the average value of residual strength of RAC is reduced by 7.89%, compared with NAC. The reason is that the shear failure of concrete consists of the failure of the interface between aggregate and mortar, with the coarse aggregate undamaged at this time. The addition of water causes the high humidity of the interface, and hence, the residual strength of recycled concrete becomes lower than that of natural concrete. The amount of free water decreases with the continued internal hydration reaction of RAC in concrete as it ages, and the leads to a reduced friction coefficient of the influenced close-to-dry sections with reduced humidity. Nevertheless, the flatness of the RAC failure plate begins to decrease, as the recycled coarse aggregate is smoothed more easily in the shear process. Therefore, at 7 days and 14 days of age, the average residual strength of RAC is reduced by 6.24% and 6.92%, respectively, compared with NAC, and this reduction is lower than that at 3 days of age. The residual strength is mainly influenced by the flatness of the failure at 28 days, and the mean value of the residual strength of RAC is 5.19% lower than that of NAC (ignoring the date of RAC with a 60% replacement ratio at 28 days of age, which is attributed to excessive error).

#### 4.2.2. The Influence of Replacement Ratio

Values of residual strength of specimens with different replacement ratios are shown in Figure 10. The overall residual strength of RAC decreases with rising replacement ratio at all ages (excluding recycled concrete with 60% replacement ratio at 28 days of age), but the rate of reduction and the underlying reason is different among samples.

Surface humidity is the decisive factor for the friction coefficient at 3 days of age, corresponding to the failure patterns as shown in Figure 2. At 14 days and 28 days, owing to surface flatness and humidity working together, the reduced rate of residual strength observed at 14 days is almost similar to that at 7 days, both being lower than the reduced rate at 3 days. At 28 days, however, the strength of coarse aggregate is the decisive factor, and the reduced rate is the smallest.

## 5. Shear Strength of RAC

### 5.1. Shear Strength Development Model

#### 5.1.1. The Effect of Age on the Model

Establishing an early strength development model for concrete is important to predict the early age strength of RAC. An exponential function model was used to fit the direct shear strength of RAC at different ages with the same replacement ratio, as shown in Figure 11a. The exponential function can be expressed as
(1)$$V_{t}/V_{28}=e^{\alpha (1-\sqrt{28/t})}$$
when *t* = 0, *V*_t_/*V*_28_^®^0;when *t* = 28, *V*_t_/*V*_28_^®^0 and ∂(*V*_t_/*V*_28_)/∂*t*^®^0.where *α* is strength coefficient of RAC related to the replacement ratio, which was obtained by fitting, as shown in Table 4. In general, the strength coefficient of RAC declines with a rising replacement ratio.


#### 5.1.2. The Effect of Replacement Ratio on the Model

The equation between strength coefficient and replacement ratio was fitted to obtain the strength coefficient of RAC at different replacement ratios, as shown in Figure 11b. The equation can be expressed as
(2)$$\alpha =-0.0006e^{r/0.1958}$$

### 5.2. Relationship between Shear Strength with Cube Compressive Strength

Direct shear strength of RAC can be determined as follows:(3)$$\tau_{u}=\frac{V_{u}}{A}$$
where *V*_u_ is the peak shear force of RAC, A is the area of the sheared surface of RAC, and A = 22,500 mm^2^. In order to investigate the relationship between cube compressive strength and direct shear strength of RAC at various ages, the ratio of axial compressive strength to direct shear strength of RAC is defined as the compression–shear coefficient *ξ.*
(4)$$\xi =\frac{\tau_{u}}{f_{\mathrm{cu}}}$$
where *τ*_u_ is the direct shear strength of RAC, and *f*_cu_ is cube compressive strength of RAC. The relationship between the replacement ratio and the compression–shear coefficient is obtained by fitting the compression–shear coefficient at different substitution rates, as shown in Figure 12.

At 3 days of age:(5)$$\xi =3.989\times 10^{-5}r+0.113$$

At 7 days of age:(6)$$\xi =6.714\times 10^{-5}r+0.100$$

At 14 days of age:(7)$$\xi =2.325\times 10^{-5}r+0.110$$

At 28 days of age:(8)$$\xi =7.941\times 10^{-5}r+0.093$$

## 6. Conclusions

In the current study, direct shear tests were conducted on RAC samples. The load–displacement curves and strength mechanical properties of specimens were obtained, as well as the failure mechanism, and the effects of replacement ratio and age on strength mechanical properties of RAC were disclosed by observing the failure pattern of specimens. The following considerations were made:(1)Interface and coarse aggregate damage is the main reason for shear plane failure. At the age of 3 days, interface damage causes specimen failure. Both interface damage and coarse aggregate damage occur at other ages.(2)The shear load–displacement curve of RAC can be divided into four stages: elasticity stage, elastoplasticity stage, plasticity stage, and stabilization stage. The elastoplasticity stage of specimens shortens with age, indicating the relative decline of concrete brittleness.(3)The peak direct shear force of RAC improves to the range of 5.6% to 14.6% at other ages, compared to NAC at ages of 3 days, 7 days, 14 days. However, at the age of 28 days, it shows a 4.2% decline, compared to NAC. Residual strength decreases with the rise in the replacement ratio. The mean residual strength of RAC is lower, ranging from 5.19% to 7.89% for NAC, and its reduction diminishes with age.(4)Age is the decisive factor in the early direct shear strength of RAC instead of the replacement ratio. Considering replacement ratio and age, a direct strength calculating formula for RAC is established. The formula for calculating the compression–shear coefficient of recycled concrete at different ages is proposed.

## Figures and Tables

**Figure 1 materials-14-02312-f001:**
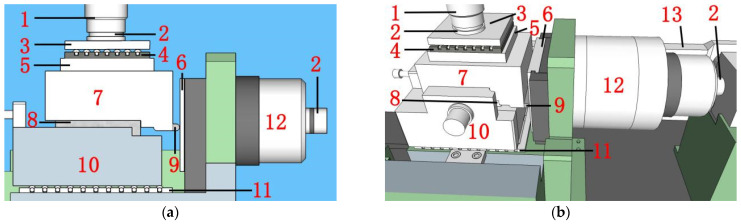
Testing device: (**a**) Front view of the testing device. (**b**) Testing device model. Note: 1-axial actuator, 2-spherical hinge, 3-platen, 4-upper rolling plate, 5-base plate, 6-restrictor plate, 7-upper shearing box, 8-specimen, 9-self-adjusting lever, 10-lower shearing box, 11-lower rolling plate, 12-lateral actuator, 13-connecting rod.

**Figure 2 materials-14-02312-f002:**
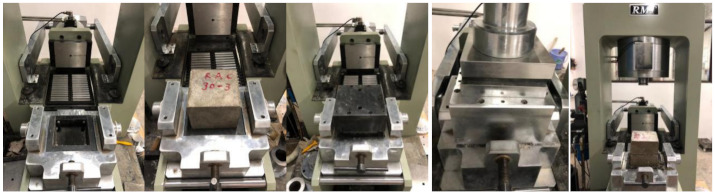
Direct shearing test: shearing device.

**Figure 3 materials-14-02312-f003:**
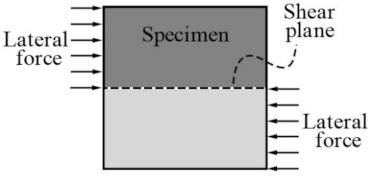
Direct shear loading diagrams.

**Figure 4 materials-14-02312-f004:**
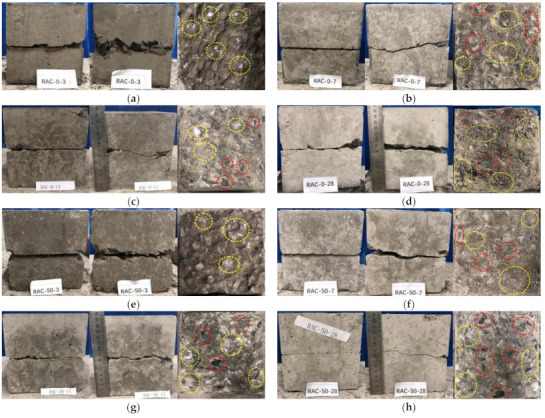
Typical failure patterns of specimens at different replacement ratios: (**a**) Failure patterns of RAC-0 at 3 days. (**b**) Failure patterns of RAC-0 at 7 days. (**c**) Failure patterns of RAC-0 at 14 days. (**d**) Failure patterns of RAC-0 at 28 days. (**e**) Failure patterns of RAC-50 at 3 days. (**f**) Failure patterns of RAC-50 at 7 days. (**g**) Failure patterns of RAC-50 at 14 days. (**h**) Failure patterns of RAC-50 at 28 days. (**i**) Failure patterns of RAC-100 at 3 days. (**j**) Failure patterns of RAC-100 at 7 days. (**k**) Failure patterns of RAC-100 at 14 days. (**l**) Failure patterns of RAC-100 at 28 days.

**Figure 5 materials-14-02312-f005:**
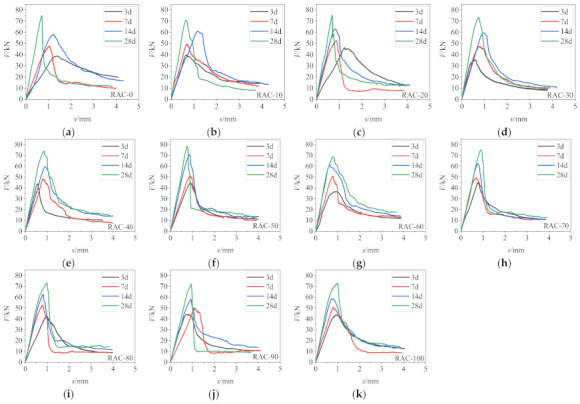
Load–displacement curves of specimens: (**a**) RAC-0. (**b**) RAC-10. (**c**) RAC-20. (**d**) RAC-30. (**e**) RAC-40. (**f**) RAC-50. (**g**) RAC-60. (**h**) RAC-70. (**i**) RAC-80. (**j**) RAC-90. (**k**) RAC-100.

**Figure 6 materials-14-02312-f006:**
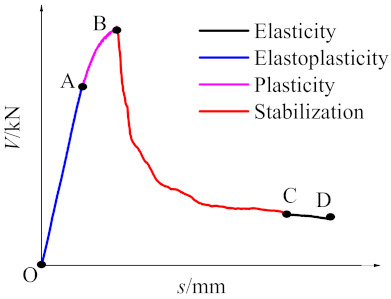
Typical load–displacement curves.

**Figure 7 materials-14-02312-f007:**
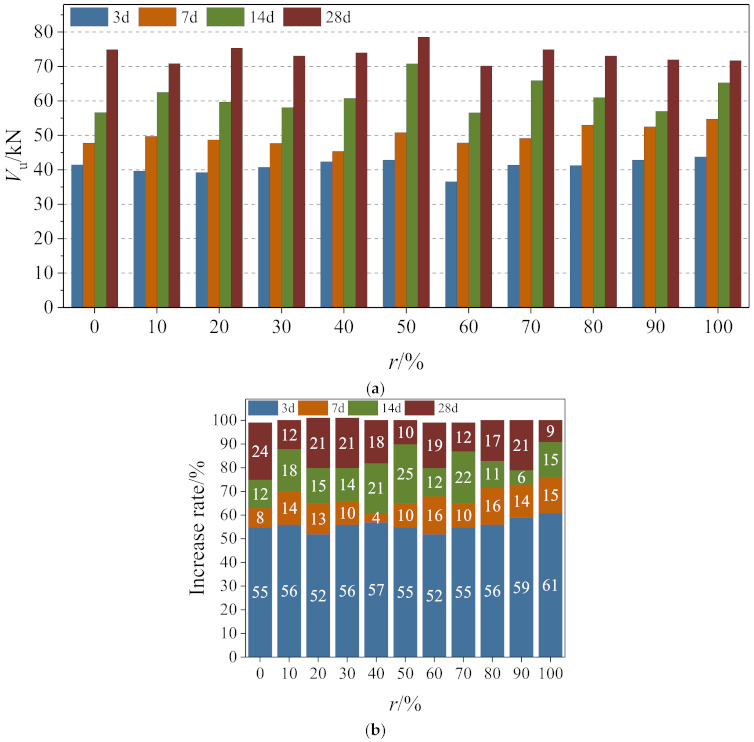
Comparison of peak shear force and increase rate of specimen: (**a**) comparison of peak shear force and (**b**) comparison of increase rate.

**Figure 8 materials-14-02312-f008:**
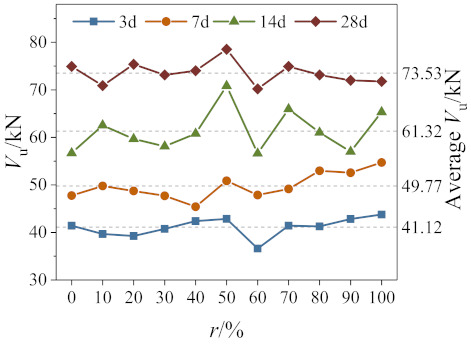
Peak shear force of RAC with different replacement ratios.

**Figure 9 materials-14-02312-f009:**
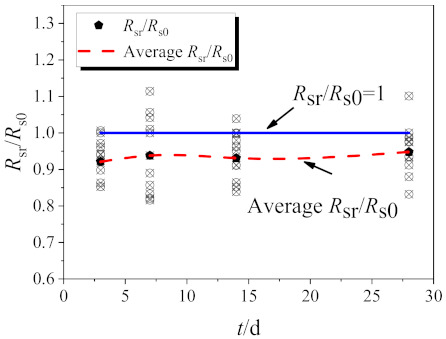
The proportion of residual strength at different ages.

**Figure 10 materials-14-02312-f010:**
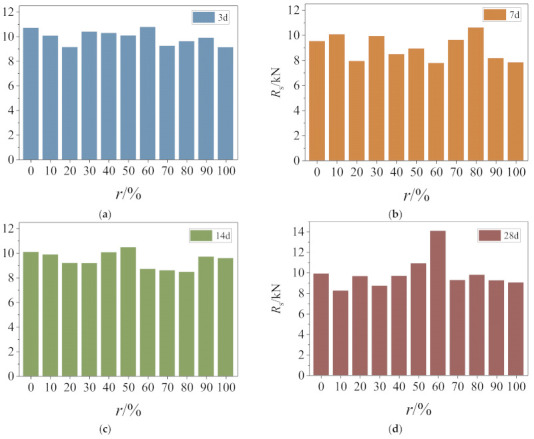
Residual strength of RAC with different replacement ratios: (**a**) 3 days. (**b**) 7 days. (**c**) 14 days. (**d**) 28 days.

**Figure 11 materials-14-02312-f011:**
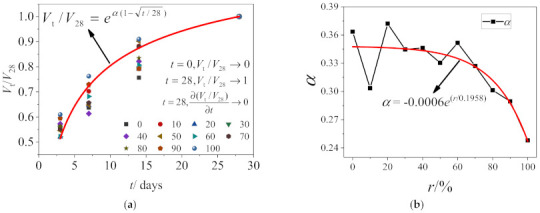
Strength development model of RAC: (**a**) the model of the fitting and (**b**) strength coefficient fitting.

**Figure 12 materials-14-02312-f012:**
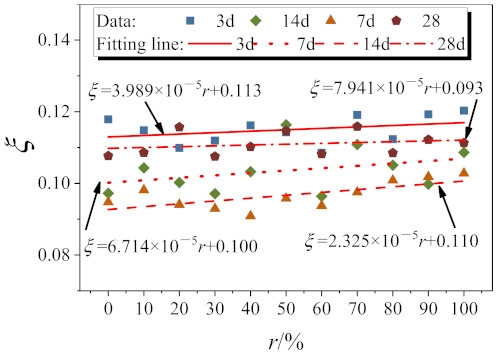
Compression–shear coefficient fitting of RAC.

**Table 1 materials-14-02312-t001:** Physical properties of the coarse aggregate.

Type	Grading (mm)	WaterContent (%)	WaterAbsorption (%)	BulkDensity (kg/m^3^)	ApparentDensity (kg/m^3^)	CrushIndex (%)
Nature coarse aggregate	5–20	0.098	0.309	1412	2714	19.89
Recycled coarse aggregate	5–20	0.715	1.68	1274	2579	22.8

**Table 2 materials-14-02312-t002:** Cement mixture design of concrete (kg/m^3^).

Ingredient	RAC-0	RAC-10	RAC-20	RAC-30	RAC-40	RAC-50	RAC-60	RAC-70	RAC-80	RAC-90	RAC-100
Cement	353.9	353.9	353.9	353.9	353.9	353.9	353.9	353.9	353.9	353.9	353.9
Water	195	197	199	201	203	205	206.9	208.9	210.9	212.9	214.9
Sand	666.4	666.4	666.4	666.4	666.4	666.4	666.4	666.4	666.4	666.4	666.4
NAC	1184.7	1066.2	947.8	829.3	710.8	592.4	473.9	355.4	236.9	118.5	0
RAC	0	118.5	236.9	355.4	473.9	592.4	710.8	829.3	947.8	1066.2	1184.7

Note: RAC-*r* for recycled concrete has an *r*% coarse aggregate replacement ratio. For example, RAC-20 for recycled concrete has a 20% coarse aggregate replacement ratio.

**Table 3 materials-14-02312-t003:** Characteristic parameters of specimens.

No.		RAC-0	RAC-10	RAC-20	RAC-30	RAC-40	RAC-50	RAC-60	RAC-70	RAC-80	RAC-90	RAC-100
3 d	*V*_u_/kN	41.43	39.68	39.25	40.76	42.4	42.86	36.62	41.42	41.28	42.83	43.77
	*R*_s_/kN	10.72	10.08	9.14	10.41	10.3	10.09	10.79	9.25	9.62	9.91	9.14
	*f*_cu_/MPa	15.61	15.33	15.83	16.17	16.18	16.63	15.02	15.45	16.29	15.93	16.21
7 d	*V*_u_/kN	47.76	49.78	48.73	47.71	45.41	50.86	47.87	49.16	52.98	52.56	54.72
	*R*_s_/kN	9.54	10.08	7.95	9.94	8.49	8.94	7.79	9.63	10.62	8.17	7.84
	*f*_cu_/MPa	22.37	22.52	23.07	22.81	22.23	23.59	22.72	22.35	23.3	22.98	23.64
14 d	*V*_u_/kN	56.69	62.55	59.67	58.12	60.79	70.84	56.62	65.96	61.03	57	65.32
	*R*_s_/kN	10.1	9.9	9.21	9.2	10.08	10.49	8.73	8.61	8.48	9.72	9.61
	*f*_cu_/MPa	25.92	26.65	26.44	26.57	26.15	27.08	26.15	26.44	25.79	25.36	26.7
28 d	*V*_u_/kN	74.93	70.88	75.38	73.13	74.03	78.53	70.2	74.93	73.13	72	71.78
	*R*_s_/kN	9.93	8.26	9.68	8.74	9.7	10.93	14.1	9.3	9.81	9.27	9.05
	*f*_cu_/MPa	30.93	29.01	28.96	30.23	29.85	30.45	28.84	28.74	29.96	28.52	28.67

Note: *V*_u_ for peak shear force, *R*_s_ for residual strength, *f*_cu_ for cube compressive strength.

**Table 4 materials-14-02312-t004:** The parameters of specimens.

ReplacementRatio/%	0	10	20	30	40	50	60	70	80	90	100
*α*	0.364	0.303	0.372	0.345	0.346	0.330	0.352	0.327	0.301	0.289	0.248

## Data Availability

The data presented in this study are available on request from the corresponding author.

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
