# Peer review of "Study on the Strength Performance of Recycled Aggregate Concrete with Different Ages under Direct Shearing"

_materials, 2021, doi:10.3390/ma14092312_

Round 1
Reviewer 1 Report
The originality and the scientific value of the subject research are good.
The research area is Study on the Strength performance of recycled aggregate concrete with different ages under direct shearing.
Experiments are performed for three different types of time (3 d, 7 d, 14 d, and 28 d).
The manuscript has the usual structure.
The experimental program is logical and interesting.
However, the limitation of the experimental program is the scope of experiment test types. It would be appropriate to perform other types of mechanical property tests at the same time, for example, compressive/tensile strength, modulus of elasticity, or fracture energy.
There is extensive research in the solved area. The introduction part needs to be reworked and extended. It is also necessary to state more information of motivation for the solved research. For a comprehensive solution to the problem, it would be appropriate to use numerical modeling.
Sucharda, O. et. al. Non-Linear Analysis of an RC Beam Without Shear Reinforcement with a Sensitivity Study of the Material Properties of Concrete. Slovak Journal of Civil Engineering 2020, 28 (1), 33-43.
Tang, W. et. al. Influence of Surface Treatment of Recycled Aggregates on Mechanical Properties and Bond Strength of Self-Compacting Concrete. Sustainability 2019, 11, 4182.
and many others.
Table 2 is on two pages.
Figure 4 is on two pages.
Table 3 is on two pages.
The results are given (interesting results). However, the manuscript must better present new knowledge and presented the benefits for further research. It is necessary to rework part of the conclusion. The results also need to be more criticized/discuss and the limitations of the experimental program determined more.
The manuscript must be revised.
Author Response
The authors thank the Reviewer for the time spent on this evaluation.

Reviewer 2 Report
My remarks and comments:
- the work was prepared quite carefully, especially the description of the test stand. In my opinion the paper is not suitable for publication in this journal, but after the (relatively large) changes it will be worth reconsidering
- the number of samples used is by no means a measure of the work's value, but it should be the significance of the results obtained. Instead of using 11 different compositions, I would suggest doing 3~5, but support the experiment with a statistical analysis of the results obtained. The abstract mentions 264 samples. If 10 compositions and 4 time points were performed, does this mean that 6 samples were tested for each point? If so, there were no standard deviations or other methods of statistical description at any stage of data presentation. Most of the data is presented without sufficient discussion and conclusions.
- citations are largely limited to authors of the same nationality as the authors, but there are strong teams around the world dealing with similar issues, maybe it will be nice to have more broad literature research?
- 134 - the table does not specify the mass or volume fractions, or even units
- 145 - the shape of the sample in the drawing of the device does not correspond to the cubic sample described above
- 172-216 - the entire chapter covers issues that do not go beyond the known engineering knowledge. As the authors wrote "It was concluded that age is the decisive factor leading to the kind of damage that occurs" - it sounds trivial
- 217-255 - the chapter is written correctly, but in my opinion it does not contain any new scientific knowledge
- 256-261 - the authors did a lot of work, but after presenting the data in the next chapter, it turns out that the trends are not clear and the composition has very little effect on the properties
- 268-289 - the presentation of the data on the radial diagram (Fig. 7a) is very disturbing - 0 should not border on 100 and certainly no lines should be drawn between them! The presentation of the data suggests that the results are almost random - have any statistical analysis been used here, are the results of single trials or just the average result presented?
- 290-309 - why the 60% samples perform the worst and the 50% samples perform the best? does the impact of sample preparation obscure the compositional changes themselves? In my opinion, a good statistical analysis can still save this article
- 319 - data presented unreadable
- 353 - I do not understand the idea of ​​connecting the measuring points with a curve. Since the samples have such a large statistical spread, it means that we will get any results between them anyway
- up to 403 - I have doubts about the results, in line with the above comments on the results statistics
- Conclusions - 410-413 is not the conclusion, but simple observation
- Conclusions - 414-416 is generally known knowledge
- Conclusions - 417-420 the described curve does not differ from the literature descriptions of shear on cement composites
- Conclusions - 421-425 is just observation, without explaining the reasons, mechanisms and physics of the process
- Conclusions - 426-429 is only a confirmation that RAC works similarly to NAC, did such a conclusion definitely require an experiment?
Author Response

(The authors gave the same response as above.)

Reviewer 3 Report
Comments
This paper studied the mechanical properties of recycled aggregate concrete. The outcome is interesting for readers. However, there are several aspects that need to be improved. The reviewer can only recommend for publication if the author satisfactorily address the following comments in the revised version.
- Why direct shear test was conducted instead of compression test? The shear failure was occurred in a plane which does not truly represent the effectiveness of RAC.
- What’s the optimal amount of RAC to replace NAC?
- The elasticity part in Fig.6 does not make sense. It should be OA rather than CD zone.
- The test setup photos should be provided.
- It seems like the replacement of NAC by RAC has no major effect on the shear strength property as plotted in Fig. 7. What’s the scientific explanation behind this result?
- Any explanation why there is an up and down pattern for 14d data in Fig. 8?
- The novelty of the study should be highlighted clearly at the end of introduction section. How this study is different from the published study in literature?
- How the outcome of this study will benefit researchers and end users? This need to be highlighted in introduction or end of conclusion.
- The background study on the properties of concrete containing waste materials should be improved. Recently waste materials were used in enhancing the properties of normal concrete [Ref: Characteristics, strength development and microstructure of cement mortar containing oil-contaminated sand] as well as 3D printed concrete [Ref: 3D-printed concrete: applications, performance, and challenges]. Suggest to include them in introduction section with proper citations to improve the background study.
I would be happy to see the revised version to understand how these comments are being addressed.
Author Response

(The authors gave the same response as above.)

Round 2
Reviewer 1 Report
Thank you for the adjustments made.
The changes made the improvement of the manuscript.
The research area and results are from the context of the manuscript can better understand.
The manuscript contains all the main information.
The manuscript can be published in the journal.
Reviewer 2 Report
Authors made some major changes in the text and referred more extensively to literature. Although the work still suffers from an overabundance of measuring points and a underflow of statistical analysis (e.g. no deviations in the data shown in Fig. 6), it is siutable to publication.
Reviewer 3 Report
I have no further comments.